# Population genetic diversity of *Schistosoma japonicum* arises from the host switching in the life cycle

Juan Long[1,☯], Zhen-Yu Xu[1,☯], Lang Ma[1], Hongying Zong[2], Jiali Wu[3], Zhipeng Zhou[4], Peijun Qian[5], Wenya Wang[5], Limeng Feng[1], Hao Yan[4], Shuying Xiao[1], Yi Yuan[3], Yuwan Hao[5], Zelin Zhu[5], Shizhu Li[5]*, Qin-Ping Zhao[1]*

1 Department of Parasitology, School of Basic Medical Sciences, Wuhan University, Wuhan, Hubei, China, 2 Demonstration Center for Experimental Basic Medicine Education, Wuhan University, Wuhan, Hubei, China, 3 Hubei Provincial Center for Disease Control and Prevention, Wuhan, Hubei, China, 4 Gong'an County Oncomelania Snail Ecological Station, National Parasitic Resource Bank of Oncomelania and Schistosoma Collection Base, Gong'an, Hubei, China, 5 National Institute of Parasitic Diseases, Chinese Center for Disease Control and Prevention, National Center for Tropical Diseases Research, WHO Collaborating Center for Tropical Diseases, Key Laboratory of Parasite and Vector Biology, Ministry of Health, Shanghai, China

☯ These authors contributed equally to this work.
* zhaoqinping@whu.edu.cn (QPZ); lisz@chinacdc.cn (SL)

## Abstract

### Background

*Schistosoma japonicum* is a multi-host parasite, including asexual amplification in snail hosts and sexual reproduction in mammalian hosts. The genetic diversity of *S. japonicum* by host switching is less understood, which could help elucidate the genetic evolution of *S. japonicum* under host pressure and provide instruction for host sampling and the infection pattern to make *S. japonicum* infection models.

### Methods

Different developmental stages of *S. japonicum* were collected and genotyped with 24 microsatellite loci, including 345 cercariae from naturally infected snails and 472 and 540 adult worms from artificially infected mice and rabbits, separately. The genetic distribution of *S. japonicum* within and among hosts by different sampling was assessed, and the genetic diversity and population structure were calculated at different population levels during host switching.

### Results

Seven cercariae were the minimum sample size to retrieve 85% of alleles for *S. japonicum* in each snail, and meanwhile, sampling parasites from 19 snails could recover 85% of the total *Na* of *S. japonicum* in all snails in this study. After infection in mice and rabbits, 8 worms per mouse and 76 worms per rabbit were the minimum samplings to retrieve 90% of alleles from each corresponding definitive host. Further, 16 mice and 2 rabbits were the least sampling size to recover 85% of the total *Na* of *S. japonicum* in all mice and rabbits,

**Data availability statement:** The dataset of microsatellite genotypes of S. japonicum is available at figshare: dx.doi.org/10.6084/m9.figshare.28448312. Other data are presented in the article and its additional files.

**Funding:** This research was funded by the National Natural Science Foundation of China (No. 81472926, 32170506 to QPZ, 32161143036, 32311540013, and 82473688 to SL), the National Key Research and Development Program (2021YFC2300800, 2021YFC2300804 to SL). The funder had no role in study design, data collection and analysis, publication decisions, or manuscript preparation.

**Competing interests:** The authors have declared that no competing interests exist.

respectively. Although no significant difference was shown for *S. japonicum* between mice and rabbits at the suprapopulation level, it is clear that the genetic diversity of worms from 20 (or 40) mice was significantly higher than that from 1 (or 2) rabbits, especially when the host sampling was not sufficiently enough. The differentiation of worms at the infrapopulation level among mice is less than among rabbits. In addition, genetic differentiation was shown between cercaria and adult worms, which was considered to relate to allele loss after host switching.

## Conclusions

The population genetic diversity of *S. japonicum* differs in different developmental stages. Host species and sampling number significantly affect the distribution pattern of alleles and the genetic structure of *S. japonicum* at the suprapopulation level.

### Author summary

Schistosomiasis is a parasitic disease transmitted in 78 countries, and over 250 million people worldwide are infected. Schistosomes are multi-host parasites, including obligatory asexual amplification in the intermediate snail hosts and sexual reproduction in the definitive mammalian hosts. *Schistosoma japonicum* was thought to be the most virulent species because of the high egg productivity and over 40 species of mammalian hosts. The genetic variation of *S. japonicum* populations among the different hosts becomes more complicated, as the selective pressure by host switching and co-evolution with different hosts would drive the parasite consequently to evolve in the infectivity, pathogenicity, and transmission pattern. However, the genetic diversity of *S. japonicum* by host switching is less understood. This investigation could help elucidate the genetic evolution of *S. japonicum* under host pressure and provide instruction for host sampling and the infective pattern to make optimal infection models. The authors found that the population genetic diversity of *S. japonicum* differs in different developmental stages. Host species and sampling number significantly affect the distribution pattern of alleles and the genetic structure of *S. japonicum* at the suprapopulation level.

## Introduction

Schistosomiasis is a parasitic disease caused by the genus *Schistosoma*, the blood flukes of trematode. It causes severe socioeconomic losses and public health problems worldwide, transmitting in 78 countries and over 250 million people were infected [1]. Schistosomes are multi-host parasites, including obligatory asexual amplification in the intermediate snail hosts and sexual reproduction in the definitive hosts, including humans and some mammalians. Definitive hosts were infected by skin contacting the freshwater containing cercariae released from snails. Of the schistosome species that can infect humans, *Schistosoma japonicum* was thought to be the most virulent species because of the high egg productivity and over 40 species of mammalian hosts [2,3]. The genetic variation of *S. japonicum* populations among the different hosts becomes complicated, as the selective pressure by host switching and co-evolution with different hosts would drive the parasite consequently to evolve in the infectivity, pathogenicity, and transmission pattern [4]. Meanwhile, a variety of anthropogenic, environmental, and ecological factors, such as water conservancy projects, conversion

of farmland to forest, expansion of wetland areas, and climate change, may change the availability of host species, population size, and infection intensity and pattern, influence genetic diversity and population structure of *S. japonicum* in hosts, consequently change the transmission mode of disease [5–8].

Previous studies showed that genetic differentiation exists among populations of *S. japonicum* in certain geographic isolations with various molecular markers [9–15]. The compatibility to snail [16], pathogenicity to host [17], and susceptibility to praziquantel [18] of *S. japonicum* differed in different regions. It proved to be the result of geographical separation, habitat isolation [10,19], co-evolution with snails [19–21], and different transmission patterns driven by definitive hosts [22–24] during a long period [19,25,26]. However, the influence on population diversity exerted by different developmental stages of schistosomes accompanied by host transfer in limited time and space is poorly understood. Factors may be related to the genetic clustering of *S. japonicum* miracidia in snails, cercariae with varied genotypes infected in mammalian hosts, and the diversity of parasites by different species of hosts [27].

The infection and growth of schistosomes in intermediate snail hosts are necessary for the transmission and epidemic of schistosomiasis. Snails can be infected with multiple miracidia and, after several rounds of asexual reproduction within the snail, *Schistosoma mansoni* cercariae with multiple genotypes can be produced in individual *Biomphalaria* snails [28,29]. Similarly, genotypes of *S. japonicum* cercariae in a snail were proved to vary from 1 to 8 when genotyping 10 cercariae per snail [27,30], even while some genotypes of cercariae in individual snails were classified as near-identical multi-locus genotypes (niMLGs) to minimise the deviation caused by somatic mutation during the asexual amplification of miracidium to cercariae [27]. Additionally, miracidia/cercariae clustering together in the same intermediate snail host have closer kinship than in different snails, leading to over-dispersed and disequilibrium distribution of cercariae among snails [27]. It strongly implies that the sample size of snail hosts could affect the allele distribution of the cercaria suprapopulation (cercariae from all snails) [27], which may further complicate the distribution pattern and population structure of schistosomes in the field, and also influence the genetic coverage of schistosomes in the artificial infection.

The definitive hosts of *S. japonicum* include humans and more than 40 species of domestic and wild mammals [24,31]. There are differences in the recovery rate and genetic diversity of adult worms in different species of definitive hosts infected with the same pool of cercariae [24,32,33]. The infectiousness of miracidium from different definitive hosts to snails was also different [34], which suggests the multi-host parasitism of *S. japonicum* might affect its population genetic structure. Wang et al. found that *S. japonicum* in pigs, cats, dogs and goats were significantly different from those in humans, water buffalo, and cattle [24], while Rudge et al. considered there is limited differentiation of *S. japonicum* among these hosts [9]. This unconformity complicates the genetic variation of *S. japonicum* among different definitive hosts, which may relate to the genetic differences deriving from cercariae of different snails [35], limited sample size, and the selective pressure under hosts. The genetic evolution and differentiation of *S. japonicum* during the host switching in the life cycle deserve deep investigation. Although *S. japonicum* collected from naturally infected hosts in the field can best approximate the actual genetic background of the schistosome population, and artificial infection in the laboratory may result in varying degrees of genotype loss [36,37], obtaining samples of *S. japonicum* through artificial infection in the laboratory is still an essential method for many studies, because of the uncontrolled environment, difficulty of sample collection, and untraceable origin in the field. It would be helpful to clarify factors that would affect the genetic coverage of *S. japonicum* during host switching by artificial infection with specific settings, such as host species, sampling size of hosts, and genetic loss.

This study investigated the genetic diversity and population structure of *S. japonicum* at different stages during host switching, including cercariae from snail hosts and adult worms from different definitive hosts like mice and rabbits. The effects of host species and sampling number on genetic diversity were also explored to understand the genetic evolution of the *S. japonicum* population under host pressure, which could also provide references for host selection and infection patterns when infected with *S. japonicum*.

## Methods

### Ethics approval and consent to participate

The procedures involving animals were carried out according to the guidelines of the Association for Assessment and Accreditation of Laboratory Animal Care International (AAALAC) and the guidelines on 'How to treat experimental animals' issued by the Ministry of Science and Technology, China. The Institutional Animal Care and Use Committee (IACUC) of the Hubei Provincial Center for Disease Control and Prevention approved the animal study protocol with the ethics review 202110137.

### Sample collection

**Snails collection.** *Oncomelania hupensis* snails were collected in the field of schistosomiasis endemic area of Gong'an County (30°12′5.27″ N, 112°16′41.67″ E), Hubei province, China. Snails were located in wastelands in a village, not a national park or other protected area or private land. Collections were carried out with the permission and assistance of the Institute of Schistosomiasis Control of Gong'an County, which supervises snail control in this county. *O. hupensis* is not an endangered or protected species. Snails were brought back to the laboratory and were washed and transferred to individual water-filled straight tubes for 3 h under light at 25°C to stimulate the emergence of cercariae for identifying *S. japonicum* infection. Finally, 40 snails were identified as infected with *S. japonicum*.

**Cercariae collection.** After cercariae were released from snails, some from each snail were transferred to a dish using a loop and washed in cercariae washing solution (5 g/L Lactalbumin hydrolysate, 6.8 g/L NaCl, 0.4 g/L KCL, 0.2 g/L CaCl$_2$, 0.2 g/L MgSO$_4$·7H$_2$O, and 1 g/L Glucose) under microscopy. After three washes, cercariae were pipetted individually with 2 μL sterilised washing solution onto a Whatman FTA card (GE Healthcare, Pittsburgh, USA). After drying, cards were stored in a sealed plastic bag in a desiccator at room temperature. At least 20 cercariae per snail were collected and stored on Whatman FTA cards individually.

**Infection and adult worms collection.** Cercariae from 40 individual infected snails were used for infection. According to our previous investigation[27], mice infection using equally mixed cercariae from individual snails can transfer more alleles from cercariae to adult worms than using cercariae from pooled snails, better keeping the consistency in the genetic structure of adult worms among mice. Then, an infection pattern using equally mixed cercariae from individual snails was used in this study. Particularly, 2-3 cercariae were collected from each individually shedding snail, and cercariae from a minimum of 15 randomly chosen snails were pooled together per mouse infection, which led to 30-40 cercariae were used to infect a laboratory mouse (Kunming mice), and 60 mice in total were infected parallelly using cercariae from 40 snails. For rabbit infection, a single laboratory rabbit (New Zealand series) was infected with 600-800 cercariae pooled equally from 40 snails (15-20 cercariae from each snail), and 3 rabbits were infected parallelly.

All mice and rabbits were sacrificed to collect worms 7 weeks post-infection. Worms were retrieved from the mesenteric veins of each mouse and rabbit by perfusion using 0.9% NaCl. Care was taken to minimise the mechanical force during perfusion and dissection by using 2

experienced technicians to process the worms as quickly as possible. Each paired worm was transferred to an individual tube for washing and marked immediately. Unpaired worms from each mouse were checked under microscopy to confirm the sex and then washed and stored in ethanol with males and females separately. Finally, 1094 worms from 48 mice (12 of 60 mice failed to be infected or died before sacrifice) and 1242 worms from 3 rabbits were collected and stored in 95% ethanol.

## Genomic DNA extraction

**From single cercaria.** A 2 mm diameter disc was removed from the centre of each sample pipetting area of the Whatman FTA card using a 2 mm Harris Micro-punch (GE Healthcare Life Sciences, Stevenage, UK). The disc was placed at the base of a 96-well, 1.2 mL U-shaped bottom plate. After one wash with 100 μL FTA purification reagent (GE Healthcare Life Sciences) for 5 mins followed by 100 μL TE buffer (10 mM Tris-Cl, 0.1 mM EDTA, pH 8.0), 14 μL of 0.1 M NaOH with 0.3 mM EDTA, pH 13.0 was added to each well and the solution left to incubate for 5 mins, after which 26 μL of 0.1 M Tris-HCl buffer, pH 7.0 was added. Then, the plate was mixed by pulsing with a vortex mixer three times for approximately 10 secs each. The solution was left at room temperature for 10 mins, pulse vortexed 10 times, and the eluate was transferred to a clean 96-well plate, stored at -20°C for further use.

**From individual adult worm.** To avoid the influence of eggs in the uterus on the genotype of the female worm, the anterior part, including the uterus, was cut off under microscopy from each adult female worm. Meanwhile, the intact male or immature female worm was used for DNA extraction. The total genomic DNA was extracted individually from each worm using a standard sodium dodecyl sulfate-proteinase K procedure [38]. Each worm was incubated and thawed in 500 μL extraction buffer containing 50 mM Tris-HCl, 50 mM EDTA, 100 mM NaCl, 0.5% SDS, and 100 μg/mL proteinase K at 55°C for 1 h with gentle mixing. DNA in solution was extracted using standard phenol/chloroform purification, followed by 3 M sodium acetate (pH 5.2) and ethanol precipitation. Pellets of DNA were washed in 70% ethanol, air-dried, resuspended in 20 μL TE (pH 8.0), and stored at -20°C for further use.

## PCR amplification, microsatellites optimisation, and genotyping

Firstly, the *28s* ribosomal DNA gene fragment was amplified from the DNA solution of a single cercaria or adult worm to confirm genomic DNA existed in each extract, using the following forward (5'- GTGGAGTTGAACTGCAAGC -3') and reverse (5'- GCTCAACAW-TAATAGTCAAACCTG -3') primers. PCRs were performed using Illustra PuReTaq Ready-To-Go PCR Beads in a 96-well plate (GE Healthcare Life Sciences). For each bead, 2 μL of FTA elution from a single larva or 0.5 μL of crude extract from a single worm were added, with 1 μL of each primer and water to final 25 μL. The amplification was implemented with the following program: 95°C for 5 mins, then followed by 40 cycles of 30 secs at 95°C, 30 secs at 60°C, and 40 secs at 72°C, finally 72°C for 7 mins. Reactions were then checked by running 5 μL on 2% agarose TAE gel.

Small-scale genotyping was done for 30 cercariae and 30 worms chosen randomly to construct optimal multiplex microsatellite loci panels for PCR amplification. Twenty-five microsatellite loci developed and proved in previous studies [16,39,40] were checked, and 15 loci that could be amplified stably for cercariae and worms in this study were selected. Finally, 24 microsatellite loci (S1 Table) were used to form 3 panels, including the 9 microsatellite loci that already showed high polymorphism in *S. japonicum* in our previous investigation [27].

Large-scale genotyping was done for hundreds of cercariae and adult worms individually. Ten cercariae were chosen randomly from each snail for 37 snails (3 of 40 snails were

not checked because of insufficient cercariae left after infection), 10 worms (5 males and 5 females) were chosen randomly from each mouse for 48 mice, and 200 worms were randomly selected from each rabbit for 3 rabbits. Microsatellite loci for each DNA sample were genotyped using the Type-it Microsatellite PCR kit (Qiagen, Manchester, UK) with the 3 multiplex panels of 24 loci. The forward primer of each pair was labelled with an appropriate fluorescent dye, including 6-FAM, HEX, TAMRA, and ROX (S1 Table). PCR reactions were prepared in a 96-well PCR plate using 6.25 μL of 2× Master Mix, 1.25 μL Q solution, 1.25 μL primers (10 μM), 2.5 μL DNA template, and 1.25 μL ddH$_2$O, then carried out in a Bio-Rad thermocycler with the program: 95°C for 5 mins, followed by 35 cycles of 30 secs at 95°C, 90 secs at 60°C for panel A, 50°C for panel B, 62°C for panel C respectively (S1 Table), and 3 mins at 72°C, after the final extension at 60°C for 45 mins. PCR reactions were checked by running 5 μL on a 3% TAE agarose gel, and successful amplifications were sent to Sangon Biotech (Shanghai, China) for fragment analysis on a 3730XL genetic analyser (Thermo, Foster City, CA, USA). Genotypes were scored using the microsatellite plugin of Geneious Prime (version 2023.0.1) [41]. All samples were amplified and genotyped twice to reduce the deviation. Finally, genotypes of 345 cercariae from 37 snails, 472 worms from 48 mice, and 540 worms from 3 rabbits were successfully generated for the following analyses. The microsatellite genotypes of *S. japonicum* were uploaded to figshare: dx.doi.org/10.6084/m9.figshare.28448312.

## Genetic diversity analyses

**Diversity of cercariae in snails by different sampling.** Firstly, the genetic diversity of each locus using all genotyped 345 cercariae was estimated to confirm the quality of all 24 loci by calculating the observed number of alleles (*Na*), effective number of alleles (*Ae*), observed heterozygosity (*Ho*), and expected heterozygosity (*He*) in GenAlEx 6.5 [42] and Polymorphism information content (PIC) in PICcalc 0.6 [43]. Deviation from Hardy-Weinberg Equilibrium (HWE) for each locus was tested with the Chi-squared test in GenAlEx, and the linkage disequilibrium test between loci was calculated in SHEsis [44]. To evaluate the relativity between genetic diversity and sampled cercariae number in each snail, *Na,* the primary index for indicating the gene abundance of a population [45,46], was further assessed for a series of cercariae subsets by randomly selecting different numbers of cercariae in individual snails, with all 24 microsatellites. Cercariae were chosen randomly at each given number, set from 1 to maximum in each snail using the base function sample in R, with 100 replicates. The relativity was also evaluated by calculating the smallest sample size of cercariae for each snail when 50% and 95% of total *Na* was set and when the *Na* reached the plateau (where *Na* has no significant difference ($P < 0.05$) with the total *Na*).

Similarly, *Na* was evaluated for each locus in different cercariae subsets by randomly selecting different numbers of snails to assess the impact of the sampling size of snails on the genetic distribution of cercariae among snails. Snails were chosen randomly at each given number of snails, set from 1 to maximum in each test using the base function sample in R, with 100 replicates. Under each replicate, the number of cercariae in each snail was set to 8, determined according to the practical feasibility and the average number of cercariae genotyped in each snail in this study. The genotypes of cercariae for the same number of snails were merged, and the *Na* was then estimated for each locus with the function *nb.alleles* in the R package hierfstat. The relativity between genetic diversity and snail number was evaluated by calculating the smallest sample size of snails when 50% and 95% of total *Na* was set, and the *Na* reached the plateau, where *Na* has no significant difference ($P < 0.05$) with the total *Na* by the corresponding locus.

**Diversity of adult worms in mice and rabbits by different sampling.** The general values of genetic diversity for 472 adult worms from 48 mice and 540 worms from 3 rabbits were first

calculated with 24 loci by *Na*, *Ae*, *Ho*, and *He,* respectively, to check the allelic consistency of adult worms among mice or rabbits. Furthermore, the relativity between *Na* and the number of adult worms in each mouse or rabbit was investigated by randomly selecting different worms of each setting number in individual definitive hosts. The number of worms was set from 1 to maximum in each test with 100 replicates. The smallest sample size of worms from each definitive host was evaluated separately when 50% and 95% of total *Na* was set, and the *Na* reached the plateau as well, where *Na* has no significant difference ($P < 0.05$) with the total *Na*. Additionally, the *Na* for each locus in different worm subsets was estimated by randomly selecting different numbers of mice or rabbits. Specifically, mice were selected randomly at each given number, setting from 1 to maximum in each test, with 100 replicates. Under each replicate, the number of worms in each mouse was set to 9, determined according to the practical feasibility and the average number of worms genotyped in this study. As for rabbits, the number of worms in each rabbit was set to 175, the average number of worms genotyped per rabbit, with the rabbit number set from 1 to 3 in each test and 3 replicates.

Since the infection with 600-800 cercariae for one rabbit had the same parasite burden as 20 mice infected, and worms from 16 mice or 2 rabbits (the same amount of cercaria infection as that of 40 mice) showed a good representation of the overall genetic level of worms in all mice or rabbits, respectively, which consequently led to the following comparison for the genetic diversity between adult worms from 20, and 40 mice and worms from 1 and 2 rabbits with *Na*, *Ae*, *Ho*, and *He*, to test the difference of genetic diversity of adult worms in different species of definitive hosts when the parasite amount of infection is the same and the sampling host was limited. We furtherly compared the difference of adult worms between 1 rabbit and 20 mice, as well as between 2 rabbits and that from 40 mice. Mice and rabbits were selected randomly at each given number, separately, with 100 replicates. The number of worms in each mouse and rabbit was set to 9 and 175 under each replicate, respectively.

**Genetic differentiation between cercariae and adult worms during host switching.** The sample number practically detected in this study far exceeded the suggested plateau value evaluated for both cercaria and adult worm analyses. It further implied that genotyped samples could represent the whole population at the corresponding level, setting as the criterion in the following comparison. Firstly, the general diversity was compared among all genotyped 472 adult worms from mice, 540 worms from rabbits, and 345 cercariae from snails with 24 loci by *Na*, *Ae*, *Ho*, and *He*.

The genetic differentiation between the adult worm population (including 472 worms from mice and 540 worms from rabbits) and the original cercarial population (345 cercariae) was evaluated by the fixation index ($F_{ST}$) and gene flow (Nm) using the function *basic.stats* in the package hierfstat of R, following Nei's method [47] and using analysis of molecular variance (AMOVA) in Arlequin ver. 3.5.2.2 [48], to investigate the change of genetic information delivered from cercariae to adult worms under host pressure. Meanwhile, the genetic differentiation of adult worms between mice and rabbits was also estimated to investigate whether host selection affects the genetics of worms from different species of definitive hosts, even when infected with the same group of cercariae.

To construct the genetic structure of three populations, we performed Bayesian multi-locus cluster analysis using STRUCTURE 2.3.4 [49] for all genotyped samples, including cercariae from snails, adult worms from mice and rabbits. The most likely number of genetic clusters (K) was estimated by setting the number of clusters as 1 to 10 and repeating each K value 10 times. A burn-in period of 10,000 iterations was used in each run, followed by 10,000 Markov Chain Monte Carlo iterations. The STRUCTURE HARVESTER [50] was used to calculate the statistic (ΔK) described by Evanno et al. [51] to determine the optimal K, which is based on the change rate of the estimated log probability between successive K values. The genetic

distances between individuals of three populations were calculated using the covariance-standardised method using GenAlEx 6.5 [43], and then Principal Coordinate Analysis (PCoA) was performed. Furthermore, the genetic associations among different populations were performed by cluster analysis based on the unweighted pair group method with arithmetic mean (UPGMA) using MEGA 11 [52], and the phylogenetic tree was then drawn accordingly.

The private alleles were investigated for populations from different species of hosts, including all genotyped cercariae from snails and adult worms from mice and rabbits, respectively, to clarify the genetic deviation among hosts during propagation. Further, the private alleles of adult worms among different sampling sizes and species of hosts infected with the same amount of cercariae were compared. Precisely, the private alleles of worms in 20 mice (40 mice) were calculated and compared with that lost in one rabbit (2 rabbits) randomly chosen, respectively, and their deviation to all hosts was also checked further to evaluate the distribution pattern of private alleles among hosts. Mice and rabbits were selected randomly at each given number, separately, with 100 replicates. The number of worms in each mouse and rabbit was set to 9 and 175 under each replicate, respectively.

### Statistical analysis

All data used in statistical analysis in this study are continuous. Differences in genetic diversity indices of loci were evaluated with a significance level of 0.05. First, the normal distribution of datasets was checked with the Sharpiro-Wilk test. A paired-samples/independent t-test was applied when the datasets met normal distribution. Otherwise, the Wilcoxon signed-rank and Mann-Whitney U-test were applied for the related/independent samples. Bonferroni correction was used to correct the P-value in multiple comparisons.

## Results

### The genetic distribution of cercariae within and among snails is significantly influenced by sampling pattern

All loci met linkage equilibrium (D' < 0.7, $r^2$ < 0.3) and deviated from Hardy-Weinberg equilibrium ($P < 0.001$). Twenty-four microsatellite loci showed high polymorphism (PIC from 0.85 to 0.96) in cercariae (S2 Table). Allele number ($Na$) varied from 16 (p58) to 63 (p6), with the mean statistics of m$Na$ at 27.46 ± 2.18. The number of effective alleles ($Ae$) varied from 6.48 (p26) to 29.20 (p6), with the mean statistics of m$Ae$ at 12.46 ± 0.95. The observed heterozygosity ($Ho$) ranged from 0.24 (p7) to 0.76 (p22), with the mean statistics of m$Ho$ at 0.46 ± 0.03. The expected heterozygosity ($He$) varied from 0.85 (p26) to 0.97 (p6), with the mean statistics of m$He$ at 0.91 ± 0.01.

In total, 659 alleles were distributed in cercariae from 37 snails by 24 loci (27.50 per locus). For cercariae infrapopulation from each snail, $Na$ varied from 4.79 to 8.38, with the mean statistics of m$Na$ in 6.79 ± 0.88; $Ae$ ranged from 3.92 to 6.13, with the mean statistics of m$Ae$ in 4.99 ± 0.61; $Ho$ varied from 0.41 to 0.53, with the mean statistics of m$Ho$ in 0.46 ± 0.04; $He$ ranged from 0.65 to 0.82, with the mean statistics of m$He$ in 0.77 ± 0.04 (S3 Table).

The deviation of $Na$ for each snail is significant among replicates when the number of cercariae is limited. With the cercariae number increased to 4, 50% of alleles can be covered, 7 cercariae were minimal to retrieve 85% of total alleles, and 6 cercariae could retrieve 85% of total $Na$ for 67% of snails (Fig 1A). The deviation of $Na$ on each locus is also significant among the same number of replicates from randomly chosen snails when snails were limited (Fig 1B). With the snail number increased to 6, 50% of alleles can be covered. When the snail number increased to 31, 95% of alleles could be covered (Fig 1B). Meanwhile, the variation of $Na$ among replicates decreased dramatically (Fig 1B). With the snail number increased to 19

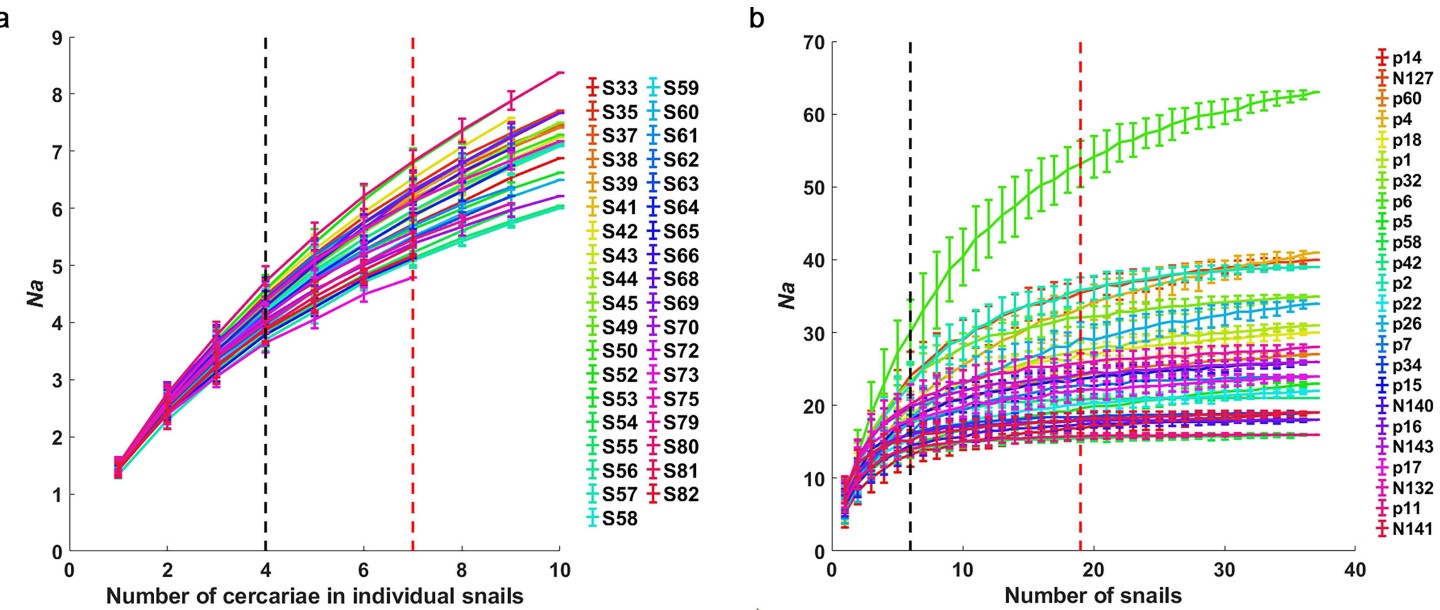

**Fig 1. The number of sampling cercariae and snails affected the coverage of alleles.** (a) Thirty-seven curves represent *Na* of cercariae in corresponding 37 snails. The red dotted line shows 85% of the total *Na* for all 37 snails, and the black dotted line shows 50%. Ten cercariae per snail were detected, except 7 cercariae in 5 individual snails because of the sample limitation. S33-S82, the serial number of the infected snail we collected. (b) Twenty-four curves represent *Na* of cercariae on corresponding 24 loci. The red dotted line shows 85% of the total *Na* for 23 loci (except p6), which reached the plateau; the black dotted line shows 50% of the total *Na*. The standard deviation of *Na* among replicates for each locus progressively decreased with the increased number of snails.

(85%), the *Na* of all loci, except p6, reached the plateau, which means the *Na* of cercariae from the specific number of snails had no significant difference with the total *Na* of cercariae from all snails by the corresponding locus. Locus p6 could not reach the plateau even while all 37 snails were sampled. It further implied that 24 loci had varied allelic frequency among snails (Fig 1B).

## The genetic differentiation by the species and sampling size of definitive hosts

For worms from individual mice, *Na* varied from 5.75 to 9.79, with the mean statistics of m*Na* in 8.06 ± 0.78; *Ae* varied from 4.35 to 7.27, with the mean statistics of m*Ae* in 5.96 ± 0.62; *Ho* varied from 0.48 to 0.74, with the mean statistics of m*Ho* in 0.62 ± 0.05; *He* varied from 0.74 to 0.85, with the mean statistics of m*He* in 0.81 ± 0.02 (S4 Table). For worms from individual rabbits, *Na* varied from 22.92 to 23.50, *Ae* varied from 12.60 to 12.70, *Ho* varied from 0.61 to 0.62, and *He* varied from 0.90 to 0.91 (S5 Table).

The non-uniform genetic distribution of worms among definitive hosts implied that the sample size of definitive hosts would affect the allele distribution of the worm suprapopulation. Generally, 727 alleles were identified for adult worms from 48 mice with 24 loci, averaging 30.30 per locus. For individual mice, 50% of alleles can be covered when worms increased to 3, and 9 worms were minimal sampling to retrieve 95% of alleles (8.06 alleles in total per mouse on average, infrapopulation level), while 8 worms (90%) were enough to reach the plateau (Fig 2A). Furthermore, 6 mice were minimal sampling to retrieve 50% of total alleles (30.29 from all mice in average, suprapopulation level), and 30 mice were necessary to retrieve 95% total alleles, while 16 mice (85%) were enough to reach the plateau (Fig 2B). As for rabbits, 741 different alleles were identified totally for adult worms from 3 rabbits, with an

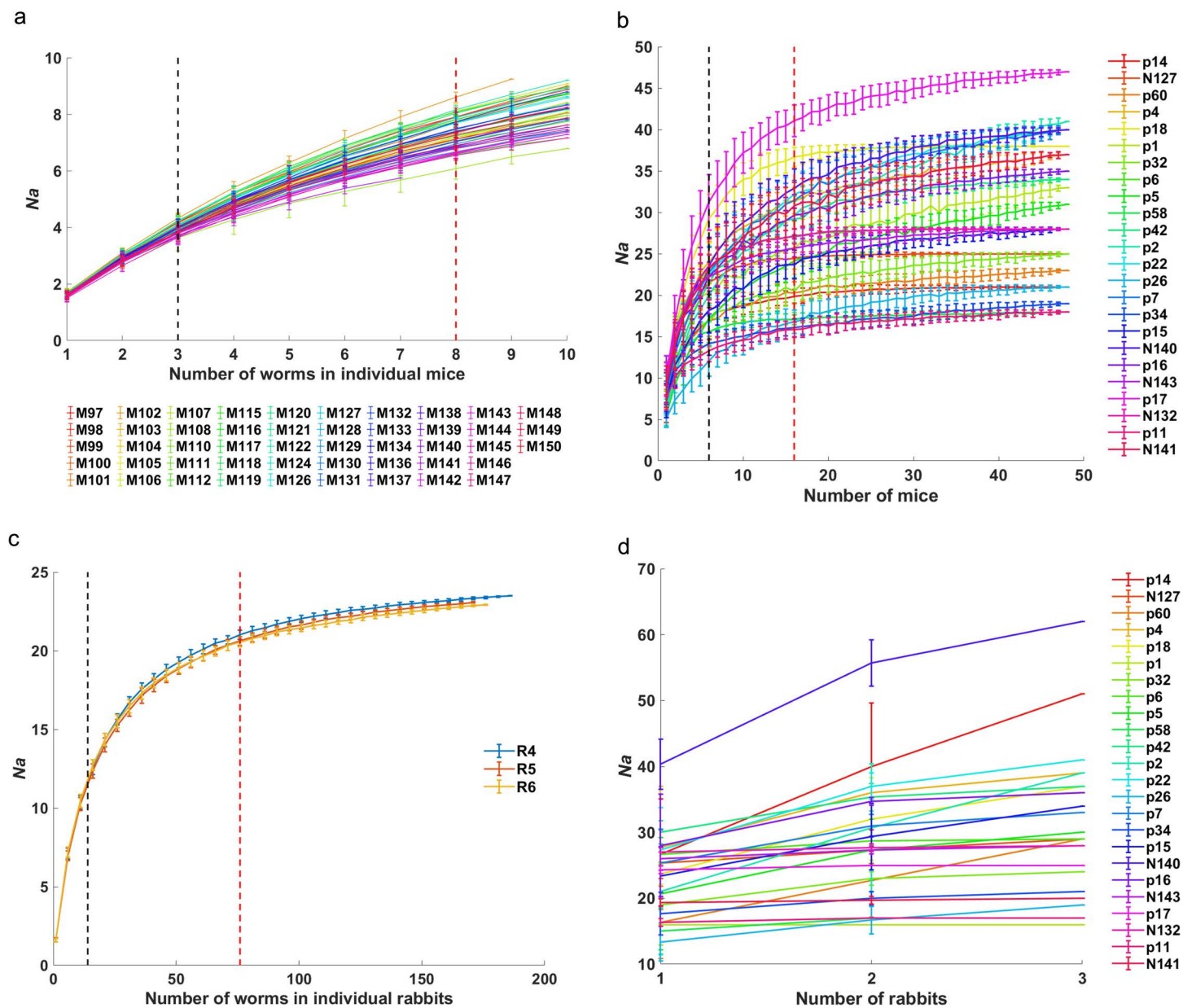

**Fig 2. The sampling size of definitive hosts and worms affects the coverage of alleles.** (a) Effect of worm sampling size in each mouse. M97-M150, the serial number of the mouse we infected. Forty-eight curves represent *Na* of worms of 48 mice. The red dotted line shows 90% of the total *Na*, and the black dotted line shows 50% of the total *Na* in each mouse. Ten worms per mouse were detected, except 7 worms in 1 mouse (M141) because of the sample limitation. (b) Effect of mouse sampling size in each locus. Twenty-four curves represent *Na* of worms on corresponding 24 loci. The red dotted line shows 85% of the total *Na* and the plateau, and the black dotted line shows 50%. (c) Effect of worm sampling size in each rabbit. R4-R6, the serial number of the rabbit we infected. Three curves represent the *Na* of worms from the corresponding 3 rabbits. The red dotted line shows 90% of the total *Na* in each rabbit, and the black dotted line shows 50% of the total *Na*. (d) Effect of rabbit sampling size in each locus. Twenty-four curves represent *Na* of worms from rabbits on corresponding 24 loci. The dotted line shows 85% of the total *Na* for all loci except p14, p60, and p2.

average of 30.90 per locus. For individual rabbits, 7 adult worms were able to retrieve 50% of alleles, and 60 adult worms were necessary to retrieve 85% of alleles (23.18 alleles in total per rabbit on average), while 76 worms (90%) were enough to reach the plateau (Fig 2C). On each locus at suprapopulation level, worms collected from 1 rabbit (33%) could retrieve 50% total alleles, and rabbits should increase to 2 (66%) when the number of retrieved alleles was set to

85% and could reach to the plateau, except p14, p60, and p2, which could not reach the plateau even all 3 rabbits were sampled (Fig 2D). It implied that the worms from one rabbit are not enough to represent the whole worm population, while 20 mice (> 16 mice) could, with the same infective dose of cercariae as one rabbit.

Furthermore, all indices (except $Ho$) of genetic diversity for worms from 20 mice showed significantly higher ($t_{(11)}$ = 4.39, $P$ = 0.010 for $Na$; $t_{(11)}$ = 3.71, $P$ = 0.003 for $Ae$; $t_{(11)}$ = 1.84, $P$ = 0.093 for $Ho$; $t_{(11)}$ = 3.92, $P$ = 0.002 for $He$) than those for worms from one rabbit. Similarly, all indices (except $He$) for worms of 40 mice were also higher ($t_{(11)}$ = 8.16, $P$ < 0.001 for $Na$; $t_{(11)}$ = 4.84, $P$ < 0.001 for $Ae$; $t_{(11)}$ = 2.75, $P$ = 0.019 for $Ho$; $t_{(11)}$ = 3.02, $P$ = 0.012 for $He$) than those from 2 rabbits (Fig 3).

## The genetic variation of *S. japonicum* during development from cercariae to adult worms by host switching

All indices (for $Na$, $Ae$, $Ho$, $He$) showed no significant difference ($t_{(46)}$ = 0.21, $P$ = 0.837 for $Na$; $t_{(46)}$ = 0.08, $P$ = 0.937 for $Ae$; $t_{(46)}$ = 0.18, $P$ = 0.856 for $Ho$; $t_{(46)}$ = 0.04, $P$ = 0.972 for $He$) between adult worms from mice and rabbits, and also no difference in the three genetic diversity parameters ($Na$, $Ae$, $He$) among cercaria and adult worm populations, except $Ho$ ($t_{(46)}$ = -3.72, $P$ < 0.001) (Fig 4A). It implied that the worms maintained a high genetic diversity originated from the cercarial population.

AMOVA showed limited differentiation between the original cercaria population and the adult worms (3% for worms from mice and 2% for worms from rabbits) (Fig 4B). The differentiation is also limited to 1% between two worm groups from mice and rabbits. Variation within individual hosts accounted for most of the genetic variation observed (58%, 59%, and 66% for variation between cercariae and worms from mice, between cercariae and worms from rabbits, and between worms from mice and rabbits, respectively). Meanwhile, variation among individual hosts accounted for 33%-39% of the difference among schistosome populations. $F_{ST}$ was 0.029 between cercariae and worms of mice and 0.025 between cercariae and

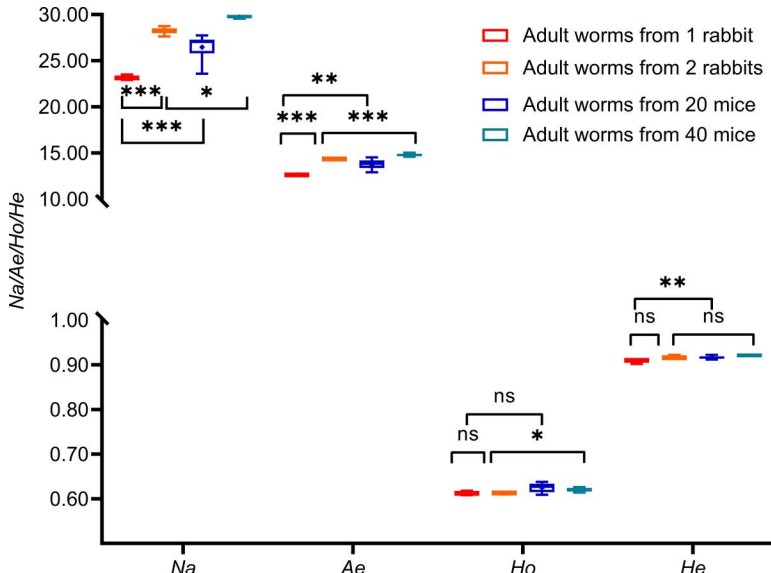

**Fig 3. Genetic diversity of worms from different mice and rabbits sampling sizes.** 1 R, one rabbit was selected; 2 R, two rabbits were selected; 20 M, 20 mice; 40 M, 40 mice.

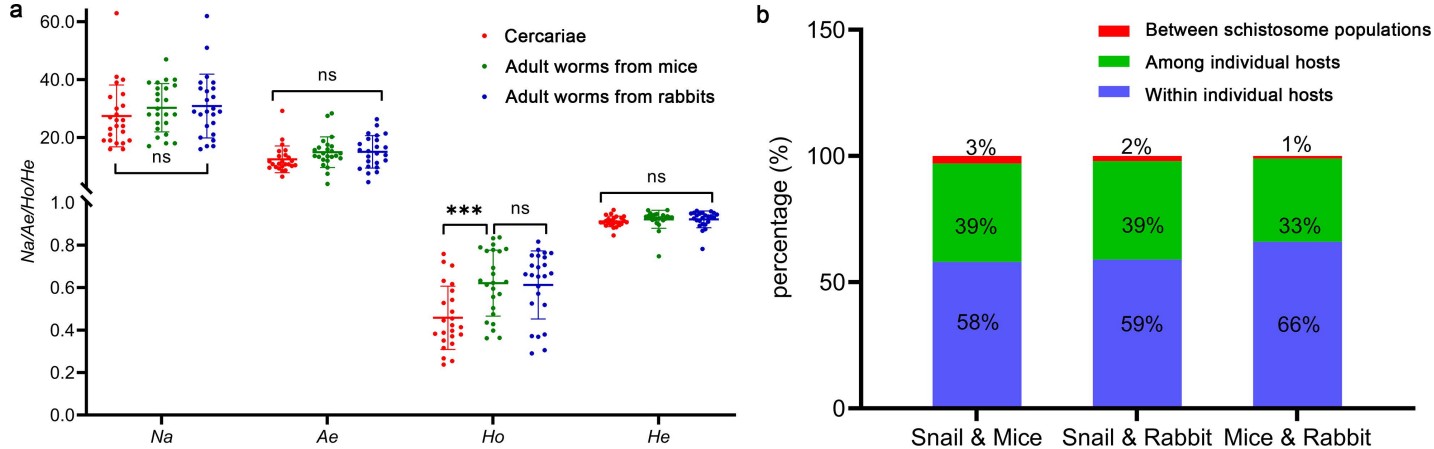

**Fig 4. Genetic differentiation of *S. japonicum* from different hosts.** (a) Genetic diversity of *S. japonicum* from different hosts. Each dot represents the genetic diversity of *S. japonicum* on each locus. The horizontal line in the middle represents the average value. (b) AMOVA analyses of different populations. "Between populations" represents molecular variance between two schistosome populations from corresponding hosts, "Among Individual hosts" represents molecular variance among individual hosts for the same schistosome population, and "Within Individual hosts" represents the molecular variance of *S. japonicum* infrapopulation within individual hosts.

worms of rabbits. Both showed similar and limited genetic differentiation, while $F_{ST}$ was 0.010 between two populations of worms from mice and rabbits. It showed that alleles on each locus could be transferred randomly from cercariae to adult worms. However, the differentiation between cercariae and the adult worms was still higher than that of adult worms between mice and rabbits.

After Bayesian clustering for all individuals, the biggest ΔK for datasets was determined at K=2 for the best representative of primary structure, suggesting that three populations could be divided into two subpopulations (Fig 5A and 5B). When K=2, populations of adult worms from mice and rabbits were grouped while the cercariae population was separated (Fig 5C). This suggested specific diversity for *S. japonicum* during different developmental stages, with a clear and distinct separation in structure.

Further, principal component analysis (PCA) was applied to investigate the possible differences between populations. The PCoA plot revealed that the samples in different hosts or developmental stages formed two clusters, one (cluster 1) comprising adult worms from mice and rabbits, and the cercarial population formed another one (cluster 2) (Fig 6A). The phylogenetic tree was constructed using UPGMA using a pairwise population matrix of Nei unbiased genetic distance. Cercariae samples were separated from adults as a separate unit, and the branches with worms from mice and rabbits were combined as the second cluster. The dendrogram showed a consistent genetic structure with that in PCoA (Fig 6B).

The private alleles were investigated separately for cercariae from snails and worms from mice or rabbits. The result showed that 26.92 and 27.83 private alleles, on average, existed in cercariae for all 24 loci but were not shared in worms from mice and rabbits, respectively (coloured in red and green in Fig 7A and S6 Table), on the premise of no base deviation allowed. Meanwhile, the lost frequency of alleles from cercariae to worms is highly varied for different loci, as from 1 for P58 to 100 for p6 that was lost in mice, and similarly, from 1 for P58 to 106 for p6 that was lost in rabbits (S6 Table). In addition, more than 70% (21.5 on average) of alleles were overlapped (coloured in blue in Fig 7A and S6 Table) that were both lost in worms of mice and rabbits but existed in cercariae. With the permission of microsatellite locus repeat deviation, the number of private alleles could decrease significantly, from 13-14

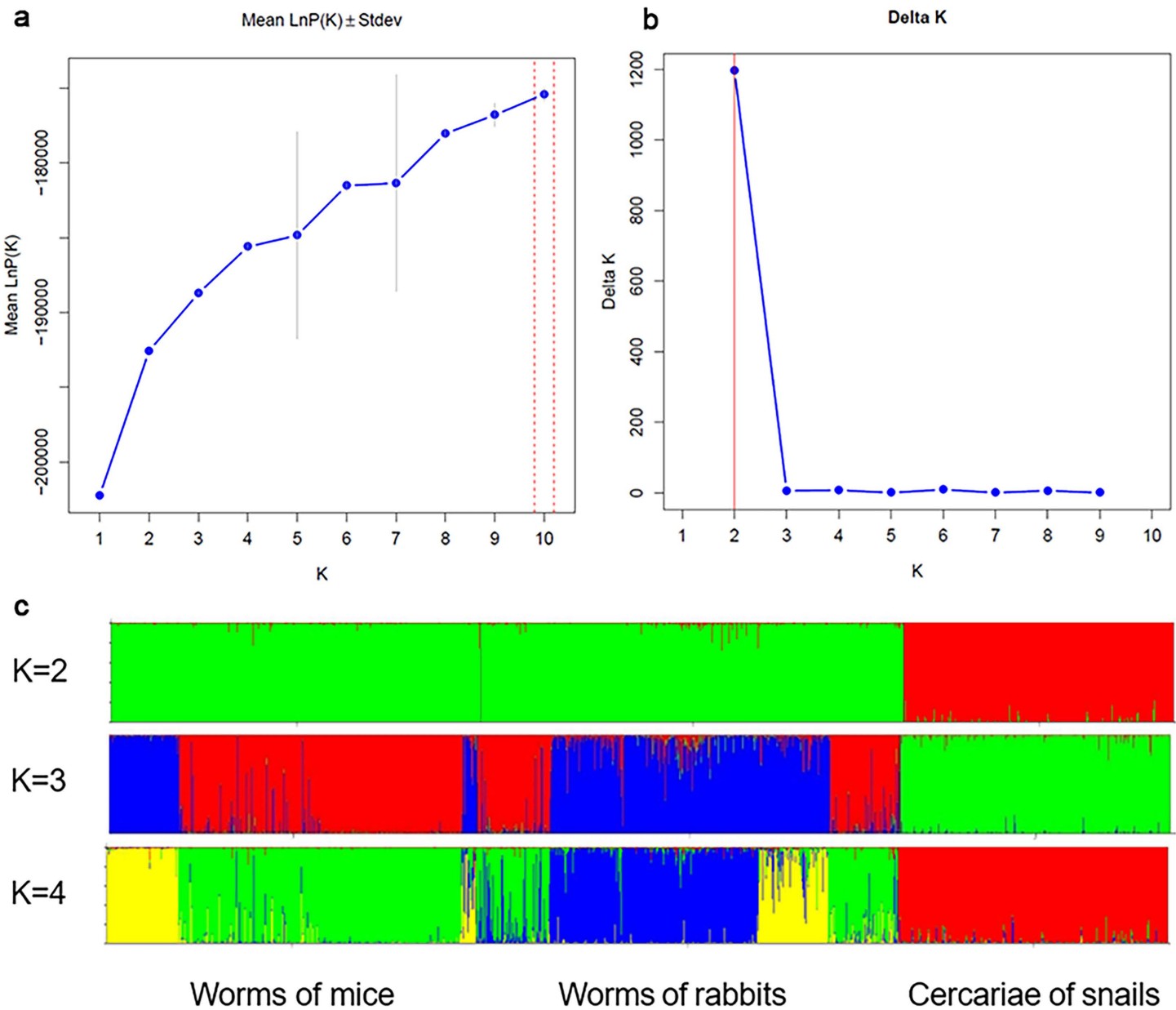

**Fig 5. Clustering for all samples using STRUCTURE.** (a) The ln likelihood [Pr (XIK)] for a given number of K over 10 times. (b) ΔK trend. (c) Population genetic structure of all samples when the estimated number of clusters K=2, 3, or 4.

private alleles with the 1 repeat deviation permitted to 8-9 alleles with the 2 repeats deviation permitted (S6 Table). Still, the proportion of overlapped private alleles did not change significantly (Fig 7B, 7C and S6 Table), 60% of private alleles both lost in mice (8/13 and 5/9) and rabbits (8/14 and 5/9), implied the similar alleles lost in different definitive hosts.

Further, the private alleles of adult worms among different sampling sizes of hosts were compared. Although the alleles lost or present from cercariae to adult worms are similar in mice and rabbits totally, the distribution of private alleles of worms among definitive hosts showed disequilibrium (S6 Table). The result showed that the private allele number of all rabbits relative to one rabbit (47.76 on average for all 24 loci, 47% of total alleles) was significantly

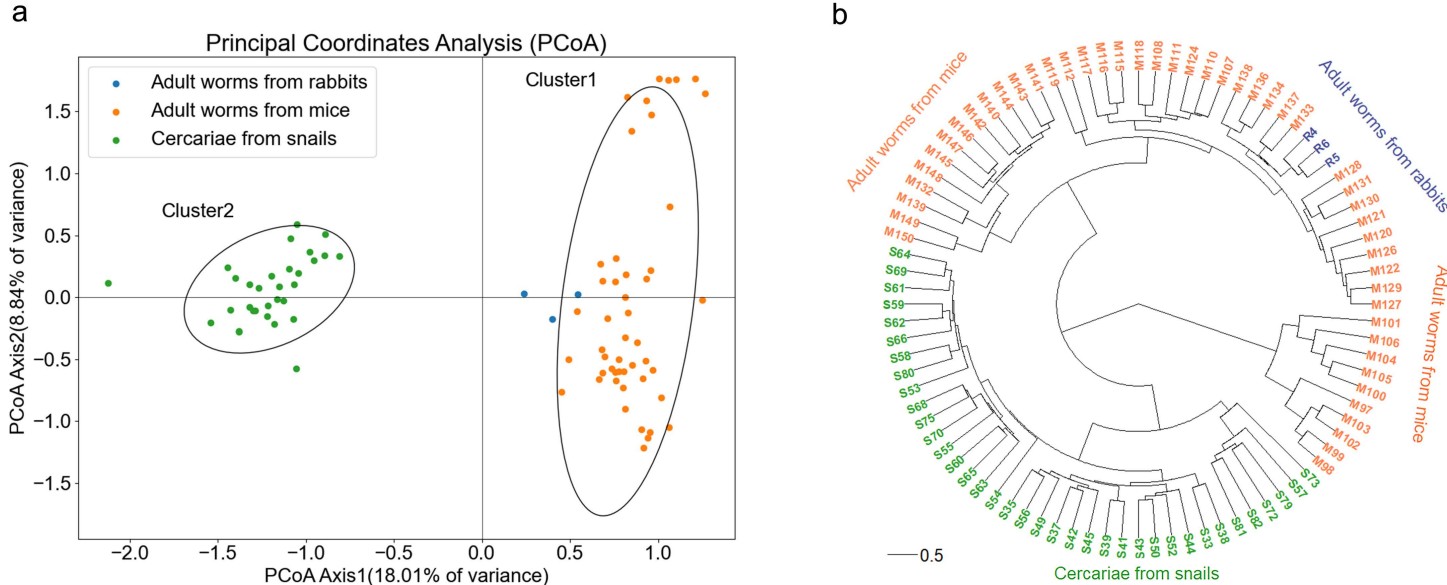

**Fig 6. Cluster analysis of *S. japonicum* from different hosts.** (a) PCoA analyses of *S. japonicum* from different hosts. (b) UPGMA tree of *S. japonicum* from different hosts based on Nei unbiased genetic distance. S33-S82, the serial number of the infected snail; M97-M150, the serial number of the mouse infected; R4-R6, the serial number of the rabbit infected.

higher than that of all mice relative to 20 mice (32.13 on average for all 24 loci, 33% of total alleles) ($t_{(46)}$ = 2.94, $P$ = 0.005), on the premise of no base deviation allowed. Similarly, the private allele number of all rabbits relative to 2 rabbits (19.81 on average for all 24 loci, 19% of total alleles) was also significantly higher than that of all mice relative to 40 mice (7.04 on average for all 24 loci, 7% of total alleles) ($t_{(46)}$ = 6.45, $P$ < 0.001) (Fig 7 and S7 Table). With the permission of base deviation, the number of private alleles could decrease significantly, from 24.41 and 4.89 for private alleles of all mice to 20 and 40 mice, 36.81 and 13.85 for all rabbits to 1 and 2 rabbits, respectively, with 1 microsatellite locus repeat deviation permitted, decreased to 18.59 and 3.76 for mice, 29.39 and 10.77 alleles for rabbits as comparison above, respectively, with the 2 repeats deviation permitted (Fig 7D and S7 Table).

## Discussion

Twenty-four microsatellite markers were used in this study. In previous related population structure studies, the number of microsatellite markers had been widely evaluated in several organisms, such as humans, plants, fruit flies, fish and sponges [53–56], with varied markers ranging from 4 to 500 even for the same species. Kovach et al. found that using a random set of 15-20 microsatellites appears to result in values that exhibit low standard deviations for diversity and differentiation indices of *S. mansoni* [57], which suggested a certain amount of microsatellite markers would take effect on the evaluation of genetic diversity of organisms including schistosomes. As the number of microsatellite markers of *S. japonicum* increased from 1 to 24, the mean value of the *Na* did not vary much (S1 Fig), while the standard deviation of *Na* decreased significantly. It could be less than 10% of the total *Na* value when using 10 markers (S1 Fig). The cut-off value used to indicate an acceptable variability in a data set is often chosen arbitrarily [58]). In the specific case of sincerity of effort, the cut-off value found in the related literature varies significantly from 5% to 20% [59,60]. The standard deviation still changes significantly when the marker number is 10 or greater in this study,

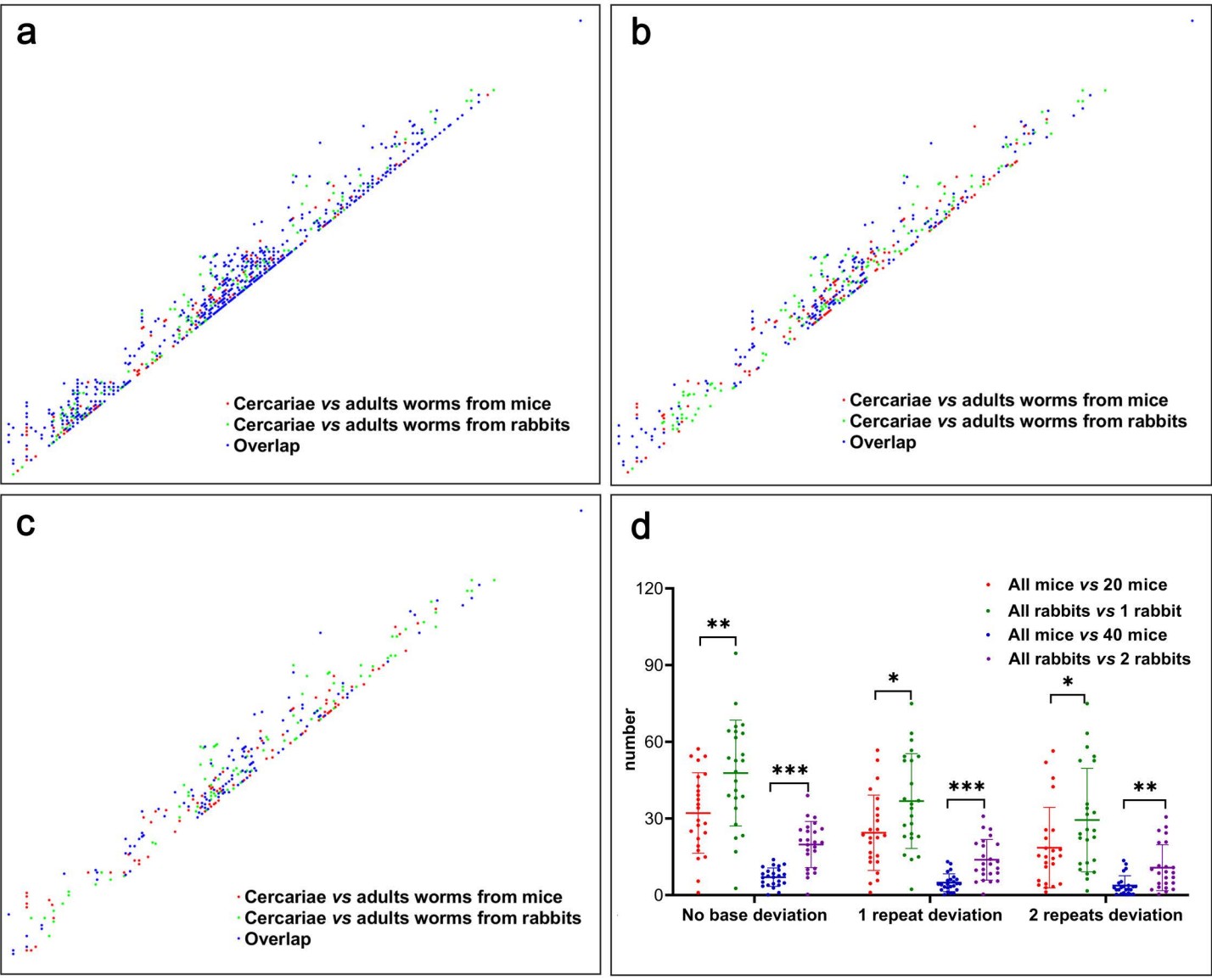

**Fig 7. Private alleles in cercariae and adult worms from different species of hosts.** (a) Private alleles of cercariae but not in worms from mice and rabbits with no base deviation. Each dot represents a private allele. "cercariae *vs* adults in mice" means the alleles present in cercariae but not in the adult worms from mice; "cercariae *vs* adults in rabbits" means the alleles present in the cercariae but not in adult worms from rabbits; "overlap" means the alleles present in cercaria but both lost in adult worms from mice and rabbits. (b) Private alleles of cercariae but not in worms from mice and rabbits with the permission of one repeat deviation of microsatellite locus. (c) Private alleles of cercariae but not in worms from mice and rabbits with the permission of two repeats deviation. (d) Number of private alleles of adult worms among different sizes of hosts with the permission of no base, one repeat, and two repeats deviations. Each dot represents the number of private alleles on a locus.

and it becomes stable after using 18 markers. Furthermore, as the coefficient of variation (CV, the ratio of mean biomass ($\mu$) to its standard deviation over time ($\sigma/\mu$), often used as a typical measure of ecosystem stability [61,62]) is 4.6% when using 18 markers, and is 9.1% when using 10 markers. Then, 18 markers are more powerful and strongly recommended (S1 Fig). Therefore, 24 microsatellite markers used in the study could help to produce more stable estimates for the genetic differentiation of *S. japonicum*.

Additionally, we analysed the cercaria population using the same 9 microsatellite markers as that used in our previous study [27]. The sample size of snails (8 cercariae in

each snail, 37 snails in total) needs to reach 7 and 30, when 50% and 95% of total *Na* was set, respectively, and the *Na* of 8 loci (except p6) could reach the plateau when 22 snails (88%) were used. When 24 loci were used, the sample size of snails still needed 6 and 31 when 50% and 95% of total *Na* were set, respectively, but the snail number decreased to 19 (85%) when the *Na* of all loci reached the plateau except p6. It showed that the sample size of snails that could reach the plateau (good representative of total alleles) reduced with the increase of loci number. The increased loci checked may expand the allele coverage in each snail and minimise the disequilibrium distribution among snails. It is practically valuable to help decrease the difficulty of sample collection from infected snails in the field. Therefore, the sampling size of cercariae in each snail should be evaluated to make sure cercariae from selected snails can cover most of the alleles of *S. japonicum* at the infrapopulation level by using a certain amount of optimal microsatellite loci, which would further influence the sampling size of snails to get the more actual genetic distribution of schistosomes at the suprapopulation level.

When all hosts were included, the adult worms in mice showed no significant genetic differentiation from adult worms in rabbits at the suprapopulation level. However, when evaluating the sample size of definite hosts that could cover most of the alleles of *S. japonicum*, the worms from 1 rabbit are not enough to represent the whole worm population, while 20 mice could, with the same infective dose of cercariae as 1 rabbit. It implied that when the sample number is limited, mice may be more representative than rabbits even under the same infection burden. Besides, genetic diversity for worms from 20 mice or 40 mice was significantly higher than that for worms from 1 rabbit or 2 rabbits, respectively, which implied that the genetic diversity of adult worms in mice was higher than that in rabbits when the sampling size of hosts was not abundantly enough and with the same amount of infection.

All of this differentiation of *S. japonicum* in genetic diversity and distribution pattern in snails, mice and rabbits led us to conclude that enough sampling size of hosts and a certain number of sampling parasites from individual hosts are preferred to a limited number of hosts and a large amount of sampling from individual hosts if genetics at the suprapopulation level was considered, which is coincident with that the genetic variability is expected to be halved in populations with very high frequencies of self-fertilization, proved by mathematical mode [63].

According to clustering analyses, *S. japonicum* showed certain genetic differentiation among different developmental stages, from cercariae to adult worms. The samples of different developmental stages from all hosts formed two clusters, one comprising adult worms from mice and rabbits, whereas the cercarial population formed another. The private alleles from different hosts were investigated and proved to be one of the causes of genetic differentiation between cercariae and worms. It showed that on the premise of no base deviation allowed, 26.90 and 27.80 private alleles, on average, existed in cercariae for all 24 loci but were not shared in worms from mice and rabbits. Among them, more than 70% (21.50 on average) were overlapped alleles that were both lost in worms of mice and rabbits, implying similar alleles lost in different definitive hosts, which is coincident with finding by Shrivastava et al. [64] that comparison of sets of cercariae from natural infections with the resultant adult worms following a single laboratory passage reflected a general loss of alleles, most likely due to founder effects and other bottlenecking processes. In addition, on the premise of no base deviation allowed, the private allele number of all rabbits relative to 1 rabbit (47% of total alleles) and 2 rabbits (19% of total alleles) was significantly higher than that of all mice relative to 20 mice (33% of total alleles) and 40 mice (7% of total alleles), respectively, which implied the private alleles of worms among mice distributed more evenly. It further suggested that

different species of definitive hosts could significantly influence the genetic distribution pattern of *S. japonicum*. The host species, host size, and sampling parasite should be considered in related studies to get more valuable population genetic information and less allele loss artificially. The population genetic change of schistosomes among different hosts also provides implications for disease epidemiology and transmission. It may promote the maintenance of genetic and phenotypic polymorphisms in infection intensity and virulence, making the effect of control strategies less predictable [65].

## Conclusions

This study investigated the genetic differentiation by the species and sampling number of hosts and the genetic variation of *S. japonicum* during development from cercariae to adult worms. The population genetic diversity of *S. japonicum* showed distinct changes during host switching. Different definitive host species and sampling numbers significantly affected the distribution of alleles and the population genetic structure of *S. japonicum*. These genetic distribution characteristics of *S. japonicum* may provide an insight into the genetic evolution of *S. japonicum* under host pressure, also a reference for host selection in laboratory artificial infection of *S. japonicum*, as well as population genetics studies of schistosomes from the field, as a certain amount of microsatellite loci, optimal host number, sampling parasite, and host species were recommended.

## Supporting information

**S1 Table.** The information on 24 microsatellite markers used in this study.
(XLSX)

**S2 Table.** Genetic diversity of cercariae for 24 microsatellite loci.
(XLSX)

**S3 Table.** Genetic diversity of cercariae from each snail.
(XLSX)

**S4 Table.** Genetic diversity of worms from each mouse.
(XLSX)

**S5 Table.** Genetic diversity of worms from each rabbit.
(XLSX)

**S6 Table.** The number of private alleles of cercariae relative to worms from mice and rabbits.
(XLSX)

**S7 Table.** The number of private alleles of adult worms among different numbers of hosts.
(XLSX)

**S1 Fig.** The number of microsatellite markers affects the deviation of genetic diversity of cercariae. The curve in the middle represents the mean value of *Na* for the different number of loci. The standard deviation of *Na* among replicates progressively decreased with the increase in the number of loci.
(TIF)

## Acknowledgements

The collection of *Oncomelania hupensis* was permitted and assisted by the Institute of Schistosomiasis Control of Gong'an County, Hubei Province, China.

## Author contributions

**Conceptualization:** Shizhu Li, Qin Ping Zhao.

**Data curation:** Juan Long, Hao Yan.

**Formal analysis:** Juan Long, Zhen-Yu Xu, Limeng Feng, Shuying Xiao.

**Funding acquisition:** Shizhu Li, Qin Ping Zhao.

**Investigation:** Juan Long, Zhen-Yu Xu, Lang Ma, Hongying Zong, Jiali Wu.

**Methodology:** Juan Long, Peijun Qian, Wenya Wang, Yi Yuan, Qin Ping Zhao.

**Project administration:** Zhipeng Zhou, Qin Ping Zhao.

**Resources:** Lang Ma, Hongying Zong, Zhipeng Zhou, Limeng Feng.

**Software:** Juan Long, Hao Yan.

**Supervision:** Shizhu Li, Qin Ping Zhao.

**Validation:** Juan Long, Zhen-Yu Xu, Yuwan Hao.

**Visualization:** Juan Long, Zelin Zhu, Qin Ping Zhao.

**Writing – original draft:** Juan Long, Zhen-Yu Xu.

**Writing – review & editing:** Juan Long, Zhen-Yu Xu, Shizhu Li, Qin Ping Zhao.

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
