## [Editor Report · Decision Letter 0]

6 Feb 2025

PNTD-D-25-00110Population genetic diversity of Schistosoma japonicum arises from the host switching in the life cyclePLOS Neglected Tropical Diseases Dear Dr. Zhao Thank you for submitting your manuscript to PLOS Neglected Tropical Diseases. After careful consideration, we feel that it has merit but does not fully meet PLOS Neglected Tropical Diseases's publication criteria as it currently stands. Therefore, we invite you to submit a revised version of the manuscript that addresses the points raised during the review process. Please submit your revised manuscript within 30 days Apr 07 2025 11:59PM. If you will need more time than this to complete your revisions, please reply to this message or contact the journal office at plosntds@plos.org. Please include the following items when submitting your revised manuscript: * A rebuttal letter that responds to each point raised by the editor and reviewer(s). You should upload this letter as a separate file labeled 'Response to Reviewers '. This file does not need to include responses to any formatting updates and technical items listed in the 'Journal Requirements' section below. * A marked-up copy of your manuscript that highlights changes made to the original version. You should upload this as a separate file labeled 'Revised Manuscript with Track Changes '. * An unmarked version of your revised paper without tracked changes. You should upload this as a separate file labeled 'Manuscript '. If you would like to make changes to your financial disclosure, competing interests statement, or data availability statement, please make these updates within the submission form at the time of resubmission. Guidelines for resubmitting your figure files are available below the reviewer comments at the end of this letter. We look forward to receiving your revised manuscript. Kind regards, Sima Zein, PhDGuest EditorPLOS Neglected Tropical Diseases Jong-Yil ChaiSection EditorPLOS Neglected Tropical Diseases

Shaden Kamhawi

co-Editor-in-Chief

Paul Brindley

co-Editor-in-Chief

**Additional Editor Comments:** The manuscript investigated the genetic diversity of Schistosoma japonicum caused by host switching during it's life cycle.

The topic of the manuscript is very interesting, however, the information in the manuscript is scattered, especially in the background section, for example: lines 74-76 give an impression that Schistosoma japonicum main hosts are snails and humans, although it was mentioned somewhere else that it can infect snails and mammals in general, nevertheless based it's results on studying microsatellites markers in population covering non-mammalian organisms (with the exception of human), those organisms were plants, fruit flies, fish and sponges (line 599)

My recommendation is that the manuscript needs to be re-organized, and clarification of why using microsatellite markers based on organisms that are not the main host of Schistosoma japonicum, ie. plants, fruit flies, fish and sponges **Journal Requirements:**

At this stage, the following Authors/Authors require contributions: Juan Long, Zhen-Yu Xu, Lang Ma, Hongying Zong, Jiali Wu, Zhipeng Zhou, Peijun Qian, Wenya Wang, Limeng Feng, Hao Yan, Shuying Xiao, Yi Yuan, Yuwan Hao, Zelin Zhu, Shizhu Li, and Qin Ping Zhao. Please ensure that the full contributions of each author are acknowledged in the "Add/Edit/Remove Authors" section of our submission form.

3) We noticed that you used the phrase 'data not shown' in the manuscript. We do not allow these references, as the PLOS data access policy requires that all data be either published with the manuscript or made available in a publicly accessible database. Please amend the supplementary material to include the referenced data or remove the references.

4) We do not publish any copyright or trademark symbols that usually accompany proprietary names, eg ©,  ®, or TM  (e.g. next to drug or reagent names). Therefore please remove all instances of trademark/copyright symbols throughout the text, including:

- TM on Lines: 165, 168, and 198.

5) Please ensure that the funders and grant numbers match between the Financial Disclosure field and the Funding Information tab in your submission form. Note that the funders must be provided in the same order in both places as well. State the initials, alongside each funding source, of each author to receive each grant. For example: "This work was supported by the National Institutes of Health (####### to AM; ###### to CJ) and the National Science Foundation (###### to AM).".

**Reviewers' comments:** **Figure resubmission:**

While revising your submission, please upload your figure files to the Preflight Analysis and Conversion Engine (PACE) digital diagnostic tool, https://pacev2.apexcovantage.com/. PACE helps ensure that figures meet PLOS requirements. To use PACE, you must first register as a user. Registration is free. Then, login and navigate to the UPLOAD tab, where you will find detailed instructions on how to use the tool. If you encounter any issues or have any questions when using PACE, please email PLOS at figures@plos.org. Please note that Supporting Information files do not need this step. If there are other versions of figure files still present in your submission file inventory at resubmission, please replace them with the PACE-processed versions. **Reproducibility:** To enhance the reproducibility of your results, we recommend that authors of applicable studies deposit laboratory protocols in protocols.io, where a protocol can be assigned its own identifier (DOI) such that it can be cited independently in the future. Additionally, PLOS ONE offers an option to publish peer-reviewed clinical study protocols. Read more information on sharing protocols at https://plos.org/protocols?utm_medium=editorial-email&utm_source=authorletters&utm_campaign=protocols

---

## [Editor Report · Decision Letter 1]

20 Feb 2025

Dear Qin Ping Zhao: 

We are pleased to inform you that your manuscript 'Population genetic diversity of Schistosoma japonicum arises from the host switching in the life cycle' has been provisionally accepted for publication in PLOS Neglected Tropical Diseases.

Best regards,

Sima Zein, PhD

Guest Editor

Jong-Yil Chai

Section Editor

Shaden Kamhawi

co-Editor-in-Chief

Paul Brindley

co-Editor-in-Chief

---

## [Editor Report · Acceptance letter]

Dear Dr. Zhao,

We are delighted to inform you that your manuscript, "Population genetic diversity of Schistosoma japonicum arises from the host switching in the life cycle," has been formally accepted for publication in PLOS Neglected Tropical Diseases.

Best regards,

Shaden Kamhawi

co-Editor-in-Chief

Paul Brindley

co-Editor-in-Chief
